# Single-atom Pt in intermetallics as an ultrastable and selective catalyst for propane dehydrogenation

Yuki Nakaya[1], Jun Hirayama[2,3], Seiji Yamazoe[2,3,4], Ken-ichi Shimizu[1,3] & Shinya Furukawa [1,3,4 ✉]

Propylene production via propane dehydrogenation (PDH) requires high reaction temperatures to obtain sufficient propylene yields, which results to prominent catalyst deactivation due to coke formation. Developing highly stable catalysts for PDH without deactivation even at high temperatures is of great interest and benefit for industry. Here, we report that single-atom Pt included in thermally stable intermetallic PtGa works as an ultrastable and selective catalyst for PDH at high temperatures. Intermetallic PtGa displays three-hold-Pt ensembles and single Pt atoms isolated by catalytically inert Ga at the surface, the former of which can be selectively blocked and disabled by Pb deposition. The PtGa-Pb/SiO$_2$ catalyst exhibits 30% conversion with 99.6% propylene selectivity at 600 °C for 96 h without lowering the performance. The single-atom Pt well catalyzes the first and second C–H activation, while effectively inhibits the third one, which minimizes the side reactions to coke and drastically improves the selectivity and stability.

[1] Institute for Catalysis, Hokkaido University, N21, W10, Sapporo 001-0021, Japan. [2] Department of Chemistry, Graduate School of Science, Tokyo Metropolitan University, Hachioji-shi, Tokyo 192-0397, Japan. [3] Elements Strategy Initiative for Catalysts and Batteries, Kyoto University, Katsura, Kyoto 615-8520, Japan. [4] Japan Science and Technology Agency, PRESTO, Chiyodaku, Tokyo 102-0076, Japan. ✉email: furukawa@cat.hokudai.ac.jp

Propylene is one of the most important building blocks for the production of a wide range of chemicals, such as polymers, resins, surfactants, dyes, and pharmaceuticals[1]. The supply of propylene has been reduced because of the recent shift in feedstock for steam crackers from oil-based naphtha to shale-based ethane. Catalytic propane dehydrogenation (PDH) using Pt- or $Cr_2O_3$-based materials is a promising on-purpose technique to satisfy the increasing global demand of propylene production[1–3]. Owing to the endothermicity, high reaction temperatures (preferably ≥600 °C) are required to obtain sufficient propylene yields. However, severe catalyst deactivation due to coke deposition and/or sintering is inevitable under such harsh conditions; therefore, the catalysts in practical use must be regenerated continuously or in short cycles. Although a number of literatures on catalytic PDH have been reported to this day, no catalyst that exhibits high catalytic activity, selectivity, and day-long stability at high temperatures (≥600 °C) has been developed to the best of our knowledge[1–5]. Developing a catalyst to meet this demanding task is of a great challenge in pure and applied chemistry.

Generally, selectivity and stability in PDH are determined by the balance between whether the product propylene desorbs or undergoes undesired side reactions, such as further C–H(C) scissions and the subsequent coke formation[6–10]. For Pt-based catalysts, Pt–Pt ensembles are known to be active for over-dehydrogenation of propylene and its hydrogenolysis[1]. The isolation of Pt atoms is a promising strategy to inhibit these undesired side reactions in PDH[11]. For instance, alloying of active main metal (mostly Pt) with a certain inactive metal (mostly typical elements such as Sn) has been a conventional approach to dilute Pt–Pt ensembles and enhance propylene selectivity and stability[1]. However, it is difficult to completely isolate Pt atoms by the conventional alloying approach. Single-atom[12–15] and single-atom alloy[11,16] catalysts are also effective tools to use isolated Pt, where active metals are atomically dispersed on an oxide support and isolated by excess amount of 11 group metal like Cu, respectively. However, it is difficult to apply them to high-temperature reactions such as PDH due to its insufficient thermal stability: Pt atoms[17] or alloy nanoparticles[11,16] without spatial separation[13] are easily aggregated to form larger nanoparticles at very high temperatures.

A possible candidate to solve this challenge is single-atom-like isolated Pt included in thermally stable intermetallic compounds. For instance, the 1:1 compound of Pt and Ga with cubic $P2_13$ space group has thermal stability ($\Delta H_f = -55.6$ kJ/mol$^{-1}$) much greater than typical random alloys ($-10 < \Delta H_f \leq 0$ kJ mol$^{-1}$) and a unique structure for this purpose as shown in Fig. 1 (refs. [18,19]). The stable (111) surface of PtGa has four different terminations displaying isolated and threefold Pt and Ga sites (hereafter signed $Pt_1$, $Ga_1$, $Pt_3$, and $Ga_3$). Here, the $Ga_3$ moiety can be regarded as a matrix to support the isolated $Pt_1$ atom; therefore, it may be possible to describe the $Pt_1$ site as "single-atom Pt". Note that there are two enantiomeric forms of PtGa unit cell (PtGa:A and PtGa:B, the former is shown in Fig. 1), because the space group $P2_13$ is chiral. In an analogous system of PdGa (space group $P2_13$), such surface termination ($Pd_3$ and $Pd_1$, which were described as trimer and single atom, respectively) has actually been observed by surface science techniques[20,21]. For the PdGa system, $Pd_3$ is known to catalyze semihydrogenation of acetylene more selectively than $Pd_1$ (ref. [20]). For PtGa in PDH, however, the $Pt_3$ site is expected to be more active for further C–H(C) scissions. Therefore, some modification that makes only $Pt_3$ sites disabled while $Pt_1$ sites available for the reaction is needed for achieving highly selective and stable PDH.

In this study, we design Pb-modified PtGa where the threefold Pt is selectively blocked by Pb deposition while the single-atom Pt

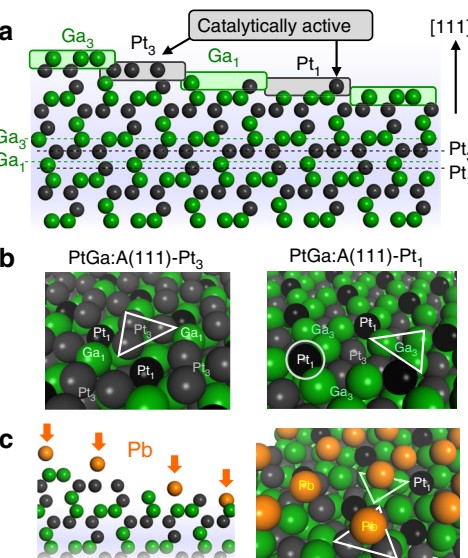

**Fig. 1 Catalyst design of single-atom Pt in PtGa. a** Four different surface terminations of PtGa:A(111) viewed along [10$\bar{1}$] direction (ball model). **b** Diagonal view of $Pt_3$ and $Pt_1$ termination (space-filling model). $Pt_1$ is highlighted with black color. **c** Catalyst design by Pb deposition to block the $Pt_3$ (and $Ga_3$) sites and to keep the $Pt_1$ sites available.

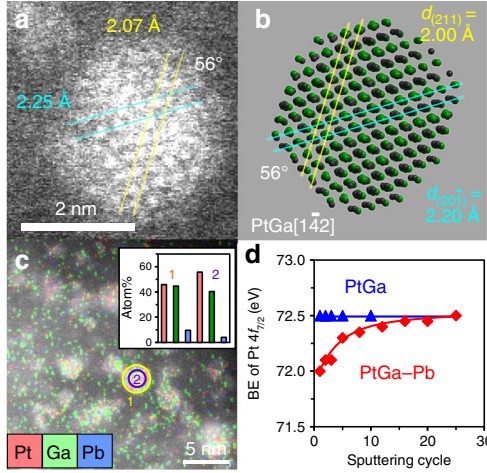

**Fig. 2 Characterization of PtGa–Pb/SiO₂. a** HAADF-STEM image of a single nanoparticle in PtGa–Pb/SiO₂ (Pt/Pb = 2). **b** Crystal structure of intermetallic PtGa viewed along [1$\bar{4}$2] direction. **c** Elemental map of Pt, Ga, and Pb acquired by EDX. Inset shows their atom% included in areas 1 (yellow circle) and 2 (purple circle), corresponding to the whole and core region of a single nanoparticle. **d** Changes in Pt $4f_{7/2}$-binding energy (BE) of PtGa/SiO₂ and PtGa–Pb/SiO₂ (Pt/Pb = 2) during sputtering cycles.

remains intact (Fig. 1c). As demonstrated later, the convex $Pt_1$ site is unfavorable geometrically and energetically for Pb deposition. We prepare SiO₂-supported PtGa and PtGa–Pb (Pt/Pb = 2) catalysts by an impregnation method (reduced by H₂ at 700 °C) and test in PDH at high temperatures (600 or 650 °C; note that it is lower than the preparation temperature). Here, we show a different type of single-atom Pt and its outstandingly high catalytic performance in PDH at high temperature.

## Results

**Characterization of catalysts.** Figure 2a shows the high-resolution high-angle annular dark field scanning transmission

electron microscopy (HAADF-STEM) image of PtGa–Pb/SiO$_2$ with a single nanoparticle. A crystal structure with interplanar distances of 2.07 and 2.25 Å and dihedral angle of 56° was observed, which agreed with those of (211) and (20$\bar{1}$) planes of intermetallic PtGa viewed along with [1$\bar{4}$2] direction[22] (Fig. 2b). The particle size distribution was narrow (mostly 1.5–3 nm, Supplementary Fig. 1) with an average of 2.8 ± 0.6 nm. The elemental map acquired by energy-dispersive X-ray (EDX) analysis showed that Pt and Ga were homogeneously distributed in each nanoparticle with approximately 1:1 ratio (Fig. 2c). Similar results of the HAADF-STEM-EDX analyses were also obtained for PtGa/SiO$_2$ (Supplementary Fig. 2, Supplementary Note 1). On the contrary, the Pb distribution in PtGa–Pb/SiO$_2$ was focused on the shell part of nanoparticles (areas 1 and 2 in Fig. 2c). Considering that the Pb content in the whole nanoparticle (area 1) is lower than those fed in the catalyst (Pt/Pb = 2), a part of Pb may present on SiO$_2$ support. X-ray photoelectron spectroscopy (XPS) analysis with Ar$^+$ sputtering revealed that the Pt$4f_{7/2}$-binding energy of PtGa–Pb was lower than that of PtGa (due to ligand effect of Pb)[23], but came close immediately after several sputtering (Fig. 2d, see Supplementary Fig. 3 for the spectra). This result strongly supports that Pb is located at the surface region of PtGa nanoparticles. We also performed X-ray adsorption fine structure (XAFS) analysis (see Supplementary Notes 2 and 3, Supplementary Figs. 4–7, and Supplementary Table 1 for XAFS analysis: Pt L$_{III}$-edge X-ray adsorption near edge spectra, extended XAFS (EXAFS) raw oscillations, EXAFS curve fits, magnitude of Fourier transform of EXAFS, and details of the curve fit). Pt–Ga scattering with 2.50 ± 0.01 Å was observed for PtGa–Pb, which is consistent finely with the interatomic distance of the nearest Pt and Ga in PtGa (2.499 Å)[22]. This result suggests that Pb atoms are not substituted into the bulk of PtGa to increase the lattice constant. Pt–Pb scattering was also observed with a small coordination number of 1.0, which indicates that the surface Pt sites are partly blocked by Pb deposition. CO pulse chemisorption experiment supported the partial coverage of surface Pt, where Pt dispersion decreased from 9.9% to 5.9% upon the Pb modification to PtGa/SiO$_2$ (Supplementary Table 2).

To obtain further information about the surface of PtGa–Pb/SiO$_2$, we then performed Fourier-transform infrared (FT-IR) spectroscopy with CO adsorption at −196 °C (Fig. 3). For PtGa/SiO$_2$, two peaks appeared at 2078 and 1885 cm$^{-1}$ at the initial stage, which are assigned to stretching vibration of CO adsorbed on Pt with on-top and threefold modes, respectively[21]. Upon the increase in CO pressure ($P_{CO}$), the threefold CO disappeared and the intensity of the on-top CO increased with an appearance of a small shoulder feature at around 2050 cm$^{-1}$. This change could be attributed to the migration of threefold CO to on-top CO on the Pt$_3$ site due to the increase of CO coverage. The new shoulder at around 2050 cm$^{-1}$ might be assigned to on-top CO adsorbed on Pt$_1$ site[21]. On the contrary, for PtGa–Pb/SiO$_2$, only a single symmetric adsorption band appeared at 2040 cm$^{-1}$ with lower intensity even at saturation coverage, which implies that the Pt$_3$ sites are blocked by Pb while the remaining Pt$_1$ sites are open for CO adsorption. We then simulated the theoretical $\nu_{C=O}$ for the suggested conformations by density functional theory (DFT) calculations (see Supplementary Fig. 8 for the detailed structures and $\nu_{C=O}$ values). The calculated $\nu_{C=O}$ values were consistent finely (on-top CO) or roughly (threefold CO) with the corresponding experimental values (Fig. 3, vertical lines), which strongly supports the assignment mentioned above. The observed trend agreed also with a relevant system of CO adsorption on PdGa:B(111) monitored by surface science techniques[21]. Only a slight red-shift in $\nu_{C=O}$ (2043 to 2037 cm$^{-1}$) was suggested when Pb was added near the Pt$_1$ site, likely because

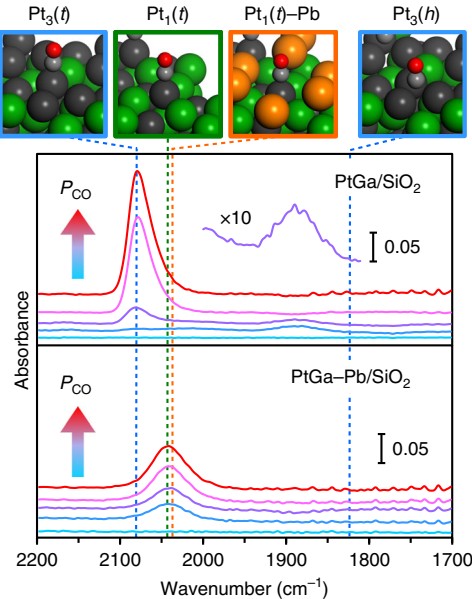

**Fig. 3 Surface characterization by FT-IR with CO adsorption.** Changes in FT-IR spectra of CO adsorbed on PtGa/SiO$_2$ and PtGa–Pb/SiO$_2$ (Pt/Pb = 2) with increase in $P_{CO}$ measured at −196 °C are shown. Vertical dashed lines indicate $\nu_{C=O}$ values calculated by DFT. The upper pictures show the optimized structures of on-top CO ($t$) on Pt$_3$, Pt$_1$, and Pt$_1$–Pb sites, and threefold CO ($h$) on Pt$_3$ hollow site.

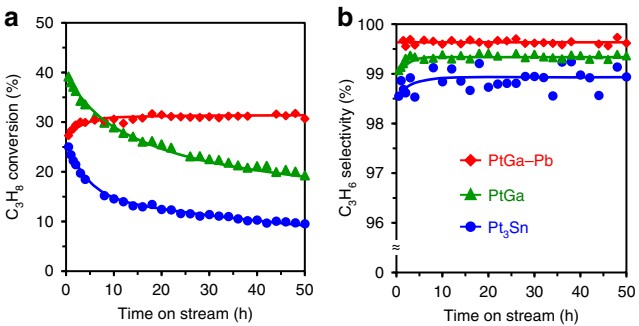

**Fig. 4 Catalytic performance in PDH. a, b** Changes in propane conversion (**a**) and propylene selectivity (**b**) in PDH catalyzed by PtGa/SiO$_2$, PtGa–Pb/SiO$_2$ (Pt/Pb = 2), and Pt$_3$Sn/SiO$_2$ are shown. Catalyst amount was adjusted so that the number of exposed Pt was identical (4.5 μmol): PtGa (9.0 mg), PtGa–Pb (Pt/Pb = 2) (15 mg), Pt$_3$Sn (3.7 mg), and diluted with quartz sand (total 1.5 g). Gas feed: C$_3$H$_8$:H$_2$:He = 3.9:5.0:40 mL min$^{-1}$. Temperature: 600 °C.

of electron-enriched Pt by the ligand effect of Pb as observed in Fig. 2d. Thus, we successfully prepared an ideal catalyst for PDH with single-atom-like isolated Pt without any Pt–Pt ensembles.

**Catalytic performance in PDH.** Next, we tested the catalytic performances of the prepared catalysts in PDH at 600 °C (Fig. 4), in which the equilibrium propylene yield in this reaction condition was approximately 60% (Supplementary Fig. 9). Although PtGa exhibited high conversion and selectivity at the initial stage (40% conv., 99.1% sel. at 0.5 h), conversion gradually decreased below half of its initial value within 50 h. Conversely, PtGa–Pb retained high conversion and excellent selectivity (>30% conv., >99.6% sel.) for 50 h even under the harsh condition. It should be

noted that almost no deactivation was observed even at 96 h (Supplementary Fig. 10). Thus, the Pb modification to PtGa significantly improved the stability and selectivity. We achieved the long-term, continuous, and highly selective propylene production in PDH at high temperatures without deactivation (>580 °C: see Supplementary Tables 3 and 4 and Supplementary Fig. 11 for comparison with literatures; deactivation rate constant was defined in Supplementary Note 4). We also tested Pt3Sn catalyst, the well-known catalyst selective for PDH[1,7], which gave lower conversion, selectivity, and stability (higher deactivation rate, Supplementary Table 5) than PtGa, highlighting the outstandingly high catalytic performance of PtGa–Pb. The spent catalysts were then analyzed by temperature-programed oxidation (TPO) and the HAADF-STEM-EDX analysis. PtGa and Pt3Sn showed coke combustion peaks in their TPO profiles, while PtGa–Pb gave no peak (Supplementary Fig. 12). This is consistent with the stability trend in Fig. 4 and suggests that the coke formation process is strictly inhibited. The HAADF-STEM-EDX analysis revealed that, despite the long-term operation (50 h) in the harsh condition, PtGa–Pb retained its small particles sizes (flesh: 2.8 ± 0.6 nm, spent: 3.0 ± 0.6 nm), intermetallic structure, and elemental distribution (Supplementary Fig. 13, Supplementary Note 5), demonstrating the high thermal stability and resistance to sintering. The stability test was also conducted at 650 °C, where PtGa–Pb retained high conversion (37–38%) for several hours and then gradually decreased to approximately 20% over 50 h (Supplementary Note 6, Supplementary Fig. 14). The gradual deactivation can be attributed to the contribution of thermal (noncatalytic) cracking[24]. This was confirmed by a control experiment using SiO2, in which small amount of C1 and C2 were formed at 650 °C, while that was negligible at 600 °C (Supplementary Fig. 15). Other bimetallic combinations that have been reported to be effective for PDH (PtSn (ref. [25]) and Pt3In (ref. [8]); see Supplementary Note 7 and Supplementary Fig. 16 for details and their X-ray diffraction (XRD) patterns, respectively) were also tested at 600 °C. However, they all showed deactivation trends similar to that of Pt3Sn (Supplementary Fig. 17, Supplementary Table 5). Considering that Pt–Ga (Pt/Ga = 3) gave higher deactivation rate and lower selectivity than PtGa, using 1:1 PtGa phase is a significant factor to develop a highly efficient catalytic system for PDH (Supplementary Fig. 17, Supplementary Table 5). When the modifier for PtGa was changed from Pb to other metals such as In or Sn, no positive effects on activity and selectivity were obtained (Supplementary Fig. 14). We also tested the recyclability of PtGa–Pb catalyst (Supplementary Note 8, Supplementary Fig. 18). The spent PtGa–Pb catalyst could be regenerated by O2 treatment to recover the original catalytic performance after some induction period, whereas some other bimetallic or trimetallic Sn-containing catalysts (Pt3Sn, PtSn, and PtGa–Sn) did not (Supplementary Fig. 19). Therefore, the combination of intermetallic PtGa and the Pb modification is suitable for stabilizing single-atom-like isolated Pt at high temperature. This is probably because (1) PtGa itself is thermodynamically stable ($\Delta H_f = -55.6$ kJ mol$^{-1}$)[18,19] and (2) the atomic radius[26] of Pb (1.80 Å) is much larger than those of Pt (1.35 Å) and Ga (1.30 Å): the diffusion of Pb into the bulk of PtGa is likely to be unfavorable even at 600 °C. Although several researchers have pointed that Ga works as a good promotor for Pt-based PDH as well as other typical element such as Sn or In (refs. [27–33]), our results indicate that the geometry and appropriate design of an active site is more significant rather than the individual chemical property of the additive element, that is, Pt should be strictly isolated. We also surveyed various Pt/Pb ratios and metal oxides as catalyst supports, which confirmed that PtGa–Pb/SiO2 (Pt/Pb = 2) was the best (Supplementary Notes 9 and 10, Supplementary Figs. 20 and 21). Ga itself has also been

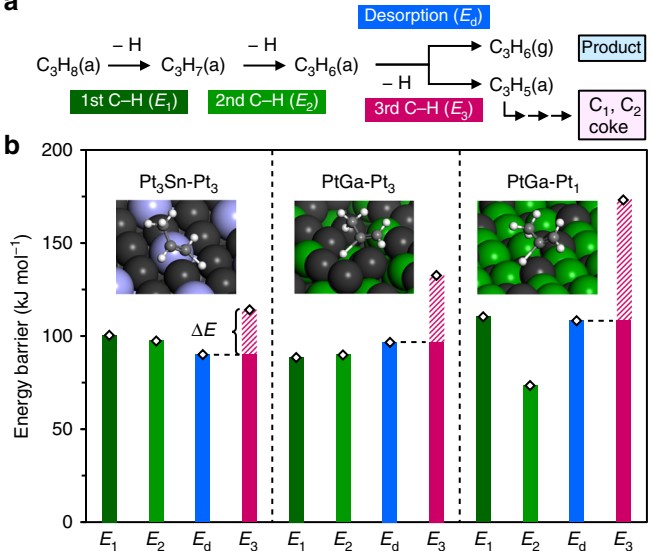

**Fig. 5 Theoretical interpretation of propylene selectivity. a** The reaction scheme of PDH to generate propylene and undesired products. **b** The energy barriers of each C–H scission and propylene desorption on Pt3Sn (111)-Pt3, PtGa:A(111)-Pt3, and PtGa:A(111)-Pt3. Inset picture shows the transition state structure in the third C–H scission for each surface. Dark green, light green, magenta, and cyan bars represent the activation energies for first ($E_1$), second ($E_2$), third ($E_3$) C–H scissions, and propylene desorption ($E_d$), respectively. Shaded parts in magenta bars correspond to $\Delta E$ ($\Delta E = E_3 - E_d$).

known to be active for PDH[34]. However, a control experiment using Ga/SiO2 at 650 °C (Supplementary Fig. 22) showed very low conversion (<3%), indicating the negligible contribution of Ga itself to the catalysis of much more active Pt-based materials.

**DFT calculations**. Finally, we conducted DFT calculations for the step-wise C–H scissions of propane to clarify the detailed property of isolated Pt for selective PDH. Figure 5 summarizes the reaction scheme of PDH and the calculated energy barrier of each step ($E_x$: $x = 1, 2, 3$, and $d$; see Supplementary Figs. 23–25, Supplementary Fig. 26, and Supplementary Table 6 for the detailed structures, energy diagram, and summarized activation energies, respectively). The adsorbed propylene (C3H6(a)) formed via the first and second C–H scissions undergoes desorption to gas phase (C3H6(g)) or further (third) C–H scission to trigger undesired side reactions[6–10]. Here, propylene selectivity depends on the difference in the two energy barriers ($\Delta E = E_3 - E_d$, shaded part in Fig. 5): the larger $\Delta E$ is, the higher the selectivity is. PtGa-Pt3 gave $\Delta E$ of 35.9 kJ mol$^{-1}$, which was slightly larger than that of Pt3Sn(111) (24.1 kJ mol$^{-1}$). Interestingly, PtGa-Pt1 having the isolated Pt showed much larger $\Delta E$ of 64.9 kJ mol$^{-1}$. This is due to the remarkably high $E_3$ (173.2 kJ mol$^{-1}$) even though $E_1$ (typically the rate-determining step of PDH) and $E_d$ did not differ significantly from those of PtGa-Pt3 and Pt3Sn-Pt3 sites. The specifically high $E_3$ could be attributed to the molecular rotation from lying 1,2-π-C3H6 to vertically standing 2-σ-C3H5 conformations occurring at the convex Pt1 site (Supplementary Fig. 25c). Because of the molecular rotation and long Pt–Pt distance between the Pt1 site and the nearest neighboring Pt3 site (3.06 Å), the hydrogen atom involved in the third C–H scission has to migrate a long distance toward the final state. The energy required for such an unfavorable path becomes significantly high. We also estimated the theoretical propylene selectivity based on

**Table 1 Theoretical and experimental $C_3H_6$ selectivity in PDH at 600 °C.**

| Theoretical simulation | | | Experimental result | |
|---|---|---|---|---|
| Surface | $\Delta E$ (kJ mol$^{-1}$)[a] | $C_3H_6$ sel. (%) | Catalyst | Initial $C_3H_6$ sel. (%)[b] |
| $Pt_3Sn$-$Pt_3$ | 24.1 | 96.5 | $Pt_3Sn/SiO_2$ | 98.6 (97.5)[c] |
| $PtGa$-$Pt_3$ | 35.9 | 99.3 | $PtGa/SiO_2$ | 99.1 |
| $PtGa$-$Pt_1$ | 64.9 | >99.9 | $PtGa$-$Pb/SiO_2$ | 99.6 |

[a]Difference between the activation energies of propylene desorption and the third C–H scission.
[b]At 0.5 h of time on stream.
[c]Catalyst amount: 15 mg.

the Arrhenius equation with $\Delta E$ (see Supplementary Note 11 for details), which are listed in Table 1 with the corresponding experimental values. The calculated values and their order were consistent with the experimental results, which demonstrates the validity of our calculation model. Thus, our calculation successfully reproduced the experimental trends in selectivity. The high propylene selectivity of $Pt_1$ sites minimizes the accumulation of coke, which leads to the outstandingly high catalyst stability. Finally, we investigated the affinity of Pb deposition to several Pt and Ga sites. Pb atoms adsorbed stably on the $Pt_3$, $Ga_3$, and concave $Pt_1$ (in PtGa-$Pt_3$ termination) sites with large adsorption energies ($-462$ to $-337$ kJ mol$^{-1}$, Supplementary Fig. 27), while could not on the convex $Pt_1$ site: the Pb atom placed on the top of the $Pt_1$ site migrated downward during structure optimization. This result indicates that the convex $Pt_1$ site is unfavorable for Pb deposition geometrically and energetically.

## Discussion

In summary, we designed and prepared the PtGa–Pb/SiO$_2$ catalyst for highly selective PDH, in which threefold hollow $Pt_3$ ensembles were successfully blocked by Pb deposition, while the single-atom-like isolated $Pt_1$ sites remained. The isolated $Pt_1$ is highly selective (99.6%) for propylene production and the catalyst is outstandingly stable for long-term operation at high temperature (96 h, 600 °C). The catalytic performance in PDH is much superior to those of the reported systems. The combination of (1) the specific crystal structure of intermetallic PtGa providing isolated Pt, (2) its thermal stability, and (3) the large atomic size of Pb enables the remarkably high selectivity and stability even in harsh conditions. The results obtained in this study provide not only a highly efficient catalytic system for alkane dehydrogenation but also significant insights for material design to isolate and stabilize active metals.

## Methods

**Materials.** SiO$_2$ (CARiACT G–6, Fuji Silysia, $S_{BET} = 673$ m$^2$ g$^{-1}$), Al$_2$O$_3$ (prepared by the calcination of boehmite [γ-AlOOH, supplied by SASOL chemicals] at 900 °C for 3 h, γ-phase), CeO$_2$ (JRC-CEO-2, $S_{BET} = 123.1$ m$^2$ g$^{-1}$), ZrO$_2$ (JRC-ZRO-6, $S_{BET} = 279.3$ m$^2$ g$^{-1}$), and TiO$_2$ (P-25, anatase). MgAl$_2$O$_4$ support was prepared by a co-precipitation method using urea as a precipitating agent. The precursors and urea were precisely weighted and dissolved together in deionized water so that urea/precursors atomic ratio was 20. Mixed aqueous solution of Mg(NO$_3$)$_2$·6H$_2$O, Al(NO$_3$)$_3$·9H$_2$O and urea was stirred overnight at 90 °C. After the precipitation, the solution was washed with water five times and dried overnight in oven at 90 °C, followed by calcination at 800 °C in the dry air for 5 h. CeZrO$_2$ and CaZrO$_3$ were prepared in a same co-precipitation method used to prepare MgAl$_2$O$_4$. Ce(NO$_3$)$_3$·6H$_2$O, Zr(NO$_3$)$_2$·2H$_2$O, and Ca(NO$_3$)$_2$·4H$_2$O were used as precursors.

**Catalyst preparation.** (1) Pt-based bimetallic catalysts were prepared by the pore-filling co-impregnation method using SiO$_2$ as the support (Pt$_3$M/SiO$_2$, and PtM/SiO$_2$, where M = Ga, In, Sn, and Pb; Pt: 3 wt%). Ga(NO$_3$)$_3$·nH$_2$O (n = 7–9), SnCl$_2$, Pb(NO$_3$)$_2$ were used as second metal precursors. The ratio of precursors was fixed at the desired ratio. Mixed aqueous solution of Pt(NH$_3$)$_2$(NO$_2$)$_2$ and second metal

was added dropwise to ground dried SiO$_2$ so that the solutions just filled the pores of the SiO$_2$. The mixture was kept in a sealed round-bottom flask overnight at room temperature, followed by quick freezing with liquid nitrogen, freeze-drying in vacuum at $-5$ °C. The resulting powder was further dried in an oven at 90 °C overnight, calcined in dry air at 400 °C for 1 h, and finally reduced by H$_2$ (0.1 MPa, 50 mL min$^{-1}$) at 700 °C for 1 h. The catalysts except Pt/SiO$_2$ were further annealed at 400 °C for 2 h under flowing H$_2$ (0.1 MPa, 50 mL min$^{-1}$) to enhance alloying without further sintering. Ga/SiO$_2$ catalyst with 5wt% loading was prepared using a similar method (reduction was carried out at 900 °C for 1 h). (2) The corresponding silica-supported trimetallic catalysts were also prepared using a similar method for PtGa/SiO$_2$ [PtGa–Pb/SiO$_2$, where Pt/Pb = 5, 2.5, 2, and 1.5; PtGa-M/SiO$_2$, where M = In and Sn, Pt/M = 2; Pt: 3 wt%]. In(NO$_3$)$_3$·8.8H$_2$O (determined by ICP-AES), SnCl$_2$, and Pb(NO$_3$)$_2$·6H$_2$O were used as third metal precursors. (3) A series of Pt–Ga bimetallic catalysts supported on various oxides (PtGa/X, where X = γ-Al$_2$O$_3$, MgAl$_2$O$_4$, CeO$_2$, CeZrO$_2$, ZrO$_2$, CaZrO$_3$, and TiO$_2$; Pt/Ga = 1; Pt: 3 wt%) was prepared by the conventional impregnation method. To prepare the precursor solution, Pt(NH$_3$)$_2$(NO$_3$)$_2$ and Ga(NO$_3$)$_3$·nH$_2$O (n = 7–9) were dissolved in an excess amount of water (ca. 25 mL of ion exchanged water per g of support). The oxide support was added to a vigorously stirred aqueous solution of the metal precursors and kept with stirring at 90 °C for 3 h. The mixture was dried using a rotary evaporator at 50 °C and further dried overnight in an oven at 90 °C. The resulting powder was treated in a similar manner for the SiO$_2$-supported alloy catalysts as mentioned above.

**Catalytic reaction.** PDH was carried out in a vertical, quartz fixed-bed reactor with 6 mm of internal diameter under an atmospheric pressure. Generally, 15 mg of catalysts diluted with quartz sand (total: 1.5 g) were charged in the reactor. For the experiments in Fig. 4, the catalyst amount was adjusted so that the number of exposed Pt was identical (4.5 μmol): PtGa (9.0 mg), PtGa–Pb (15 mg), Pt$_3$Sn (3.7 mg). Prior to the catalytic test, the catalyst was prereduced under flowing H$_2$ at 650 °C and held at 650 °C for 0.5 h. After the pretreatment, the temperature was kept at 650 °C or decreased to 600 °C, followed by feeding reactant gas mixture; C$_3$H$_8$:H$_2$:He = 3.9:5:40, a total of 48.9 mL min$^{-1}$ (WHSV = 30.7 h$^{-1}$). The resulting product gas was analyzed by online thermal conductivity detector (TCD) gas chromatograph (Shimadzu GC-8A with a column of Unipak S, GL Science) equipped downstream. For all the catalysts, C$_3$H$_8$, C$_2$H$_4$, C$_2$H$_6$, and CH$_4$ were detected as reaction products. The C$_3$H$_8$ conversion, C$_3$H$_6$ selectivity, C$_3$H$_8$ yield, and material balance were defined by Eqs. (1)–(4), respectively. Material balance typically ranged between 95% and 105% for all the reactions.

$$C_3H_8 \text{ conversion (\%)} = \frac{[C_3H_8]_{inlet} - [C_3H_8]_{outlet}}{[C_3H_8]_{inlet}} \times 100 \quad (1)$$

$$C_3H_6 \text{ selectivity (\%)} = \frac{[C_3H_6]}{[C_3H_6] + \frac{2}{3}[C_2H_6] + \frac{2}{3}[C_3H_4] + \frac{1}{3}[CH_4]} \times 100 \quad (2)$$

$$C_3H_6 \text{ yield (\%)} = \frac{[C_3H_6]_{outlet}}{[C_3H_8]_{inlet}} \times 100 \quad (3)$$

$$\text{Material balance (\%)} = \frac{[C_3H_8]_{outlet} + [C_3H_6] + \frac{2}{3}[C_2H_6] + \frac{2}{3}[C_2H_4] + \frac{1}{3}[CH_4]}{[C_3H_8]_{inlet}} \times 100 \quad (4)$$

**Characterization.** XRD patterns of the Pt-based catalysts were obtained using a MiniFlex 700+D/teX Ultra (X-ray source: Cu Kα radiation). HAADF-STEM analysis was performed by an FEI Titan G2 or a JEOL JEM-ARM200 M microscope with an EDX detector. The volume averaged particle size in a TEM image ($d_{TEM}$) was obtained by the following equation:

$$d_{TEM} = \frac{\sum_i n_i d_i^4}{\sum_i n_i d_i^3}, \quad (5)$$

where $n_i$ and $d_i$ indicate the number of particles (having the size of $d_i$) and the particle diameter, respectively.

Pt dispersion in the catalysts (percentage of exposed Pt to the total amount of Pt) was measured by chemisorption of CO at room temperature. Prior to chemisorption, the catalyst (50 mg) was treated by 5% H$_2$/Ar (40 mL min$^{-1}$) at 300 °C for 0.5 h, followed by cooling to room temperature with an He purge (40 mL min$^{-1}$) to remove chemisorbed hydrogen. We introduced a pulse of 10% CO/He into the reactor and quantified the CO passed through the catalyst bed using a TCD detector. This pulse measurement was repeated until no more CO was adsorbed. We estimated the amount of chemisorbed CO assuming a 1:1 stoichiometry for CO chemisorption on a surface Pt atom.

XPS study was conducted using a JEOL JPS-9010MC spectrometer (X-ray source: Mg-Kα radiation). The catalysts were treated by flowing H$_2$ at 650 °C for 0.5 h in a quartz reactor, followed by transferring into the spectrometer in air. The surface of the catalyst was sputtered by Ar$^+$ (voltage: 400 V, rate: 20%, time: 1 s, at each cycle) for the depth analysis. Calibration of the binding energy was performed with the Si 2p emission of the SiO$_2$ support (103.9 eV).

FT-IR spectra of adsorbed CO were obtained with a JASCO FTIR-4100 spectrometer with a TGS detector in the transmission mode (resolution $4\ \text{cm}^{-1}$) under a dynamic condition. Prior to CO chemisorption, 50 mg of the catalyst was pressed into a pellet (diameter of 20 mm) and placed in a quartz cell equipped with $CaF_2$ windows and a Dewar vessel, followed by reduction under a flowing $H_2$ at 550 °C for 1 h. The reduced sample was then kept in vacuum at 550 °C for 1 h, then the cell was cooled to ca −196 °C by liquid nitrogen. The sample was exposed to a pulse of low-pressure CO, and then evacuated in vacuum to remove the gaseous CO and concentrated CO on the catalyst. This CO exposure was repeated several times until the CO saturation coverage.

TPO experiment was performed to quantify the amount of coke deposited on the spent catalysts after 20 h of PDH at 600 °C (15 mg of the catalyst without quartz sand). The spent catalyst (10 mg) placed in a quartz tube reactor was treated under flowing He $(40\ \text{mL min}^{-1})$ at 150 °C for 30 min, followed by cooling to room temperature. Then, the catalyst bed temperature was increased (25–900 °C, ramping rate: $5\ °\text{C min}^{-1}$) under flowing $O_2$/He (50%, $40\ \text{mL min}^{-1}$). The amount of $CO_2$ in the outlet gas was quantified by an online mass spectrometer.

XAFS spectra of the prepared catalysts were collected at the BL01B1 beamline of SPring-8, Japan Synchrotron Radiation Research Institute (JASRI) using an Si(111) double-crystal as a monochromator. Prior to the measurement, the catalyst was pelletized (ca. 150 mg with a diameter of 10 mm) and pretreated by $H_2$ at 650 °C for 0.5 h in a quartz tube. After the pretreatment, the quartz tube containing the reduced pellet was sealed and transferred into an Ar grove box ($O_2$: <0.1 ppm) without exposing to air. The pellet was sealed in a plastic film bag (Barrier Nylon) together with an oxygen absorber (ISO A500-HS: Fe powder). The Pt $L_{III}$- and Ga K-edges XAFS spectra were recorded in a transmission mode at room temperature. Athena and Artemis software ver. 0.9.25 implemented in the Demeter package[35] was used for the analysis of the obtained XAFS spectra. Fourier-transform of the Pt $L_{III}$-edge EXAFS oscillation was obtained in the $k$ range of $3-16\ \text{Å}^{-1}$. The back Fourier-transform obtained in the $R$ range of $1.5-3.5\ \text{Å}$ was used for curve-fitting. FEFF8 was used for the calculation of the back-scattering amplitude and phase shift functions[36]. We defined the $R$-factor ($R^2$) for curve-fitting as follows:

$$R^2 = \Sigma_i\left\{k^3\chi_i^{exp}(k) - k^3\chi_i^{fit}(k)\right\}^2 \text{per } \Sigma_i\left\{k^3\chi_i^{exp}(k)\right\}^2. \qquad (6)$$

**Computational details**. DFT calculations were performed by using the CASTEP code[37]. We used Vanderbilt-type ultrasoft pseudopotentials[38] and the revised version of Perdew−Burke−Ernzerhof exchange-correlation functional[39,40] based on the generalized gradient approximation. A cut-off energy of 370 eV was used for the plane-wave basis set. A $k$-point mesh with a spacing of $0.04\ \text{Å}^{-1}$ generated by the Monkhorst−Pack scheme[41] was used to sample the Brillouin zone. In this study, the PtGa:A(111) and Pt$_3$Sn(111) planes were considered as the standard active surfaces for PDH. The supercell structure was constructed using a ($2 \times 2$) unit cell slab with six atomic layers and a vacuum spacing of 15 Å. We performed geometry optimizations on the supercell structures using a Fermi smearing of 0.1 eV, the OBS method for dispersion correlations, and the following convergence criteria: (1) self-consistent field tolerance: $1.0 \times 10^{-6}$ eV per atom; (2) energy tolerance: $1.0 \times 10^{-5}$ eV per atom; (3) maximum force tolerance of $0.05\ \text{eV Å}^{-1}$, and (4) maximum displacement tolerance of $1.0 \times 10^{-3}$ Å. Transition state search was carried out based on the complete linear synchronous transit/quadratic synchronous transit method[42,43] with the tolerance for all root-mean-square forces on an atom of $0.10\ \text{eV Å}^{-1}$.

## Data availability
The data that support the findings of this study are available from the corresponding author upon reasonable request.

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

## Acknowledgements

This work was supported by JSPS KAKENHI (Grant Numbers 17H01341, 17H04965, and 20H02517), MEXT project Element Strategy Initiative (JPMXP0112101003), JST CREST (JPMJCR17J3), and JST PRESTO (JPMJPR19T7). The XAFS analysis was performed with the approval of JASRI (No. 2019B1620 and 2019B1469). We appreciate the technical staffs of the faculty of engineering, Hokkaido University and of Research Institute for Electronic Science, Hokkaido University for help with HAADF-STEM observation. Computation time was provided by the supercomputer systems in Institute for Chemical Research, Kyoto University.

## Author contributions

S.F. and Y.N. design the research and co-wrote the manuscript in discussion. Y.N. performed most of the experimetal works. J.H. and S.Y. carried out the XAFS analysis. S.F. conducted the computational studies. S.F., Y.N., J.H., S.Y., and K.S. discussed the data and commented on the manuscript.

## Competing interests

The authors declare no competing interests.
