## [Peer Review File · Nature Communications]

REVIEWER COMMENTS

Reviewer #1 (Remarks to the Author):

In the present work, the authors reported an intermetallic PtGa catalyst, of which surface is selectively blocked by Pb deposition. The catalyst showed remarkably high selectivity and stability (coke resistance) during the PDH reaction. I believe that the study was carried out with very high scientific rigor. The structural characterization of catalysts, theoretical simulations, and reaction results (including the comparison of many reported bimetallic catalysts) were all of high quality. I believe that the paper can be published in Nature Communications if the authors can address the following points.

1. The authors compared several bimetallic catalysts in PDH. However, only the reactant conversion and propylene selectivity were compared. In my view point, most bimetallic and trimetallic catalysts suffer from dealloying due to the repeated redox cycles during a regeneration process. I strongly recommend the authors to additionally compare the recyclability of the prepared catalysts under a fixed regeneration condition. I believe that the present Pb-PdGa catalyst would be superior to other catalysts.
2. The authors seemed to use carefully controlled regeneration conditions; oxidation under 5% O₂/He at 550°C followed by H₂ reduction at 650°C. Is there specific reasons for choosing such a condition? What will happen if the coked catalyst is simply calcined in air (20% O₂) at the reaction temperature (more practical conditions)?
3. I am not sure whether the catalyst can be described as a “single-atom” catalyst. The Pb-modified PtGa catalyst does not have exposed Pt ensembles due to the site blocking by Ga and Pb. However, “single-atom catalysts” generally mean the metal atoms or ions isolated within a “support” matrix.

Reviewer #2 (Remarks to the Author):

Review of Single-Atom Pt in Intermetallics: An Ultrastable and Selective Catalyst for Propane Dehydrogenation by Nakaya et al.

The authors present a comprehensive study of a novel Pb-doped PtGa single Pt atom catalyst that performs exceptionally well for the selective dehydrogenation of propane to propylene. This PtGa-Pb/SiO₂ catalyst exhibits very high selectivity towards propylene and activity comparable to the more well-known Pt₃Sn catalyst. However, more pertinently, PtGa-Pb/SiO₂ shows no signs of deactivation due to coke formation after continuous runs of up to 96 hrs whereas other existing catalysts tend to deactivate to ~half their initial rate within a day or two precluding their practical use. The authors investigation concludes that Pb selectively blocks more reactive Pt₃ surface sites but not the more selective single Pt sites that are thought to be responsible for the observed catalytic performance. Further DFT calculations reveal that unblocked single Pt sites exhibit comparable barriers for C-H activation in C₃H₈ and C₃H₇ though a very high barrier for further dehydrogenation that far exceeds that for propylene desorption. It is my opinion that the authors have presented an excellent contribution that is of the quality and novelty expected from an article in Nature Communications.

Therefore I recommend that, subject to consideration of the minor comments below and some wordsmithing by the editorial team, this manuscript should be accepted for publication.

Minor Comments:

pg 3 line 6: "Owing to the endothermicity and limited equilibrium propane conversion, high temperatures are requires..." – limited equilibrium propane conversion does not make sense.

pg 3 line 16: "For Pt-based catalysts, Pt–Pt ensembles are known to be active for deep C–H(C) activation and their hydrogenolysis." – what is "deep" C-H activation? Is it C-H scission from lower valency C atoms? The term is used in several places and is not, as far as I'm aware, commonplace in the C-H activation literature.

Figure 2d: A more appropriate label for the x-axis would be "Sputtering Cycle".

pg 6 line 20: "CO pulse chemisorption experiment supported the partial coverage of surface Pt, where Pt dispersion decreased from 9.9% to 5.9% upon the Pb modification to PtGa/SiO₂" – this isn't terribly clear though after checking Table S11, it seems that as Pt/Pb decreases, the Pt % dispersion also decreases. The text should be revised to convey this. Also, the "Pt dispersion" should be defined.

pg 8 line 19: "Although PtGa exhibited high conversion and selectivity at the initial stage (40% conv. 99.1% sel. at 0.5 h), conversion gradually decreased below half within 50 h." – half of it's initial value, rather than absolutely half.

pg 11 Figure 5 and discussion: units of eV are used in the figure and mostly kJ/mol in the text/Table 1. This should be made uniform.

pg 11 line 15: "The displacement of the hydrogen atom to be eliminated becomes larger due to the absence of closely neighboring Pt, which significantly destabilizes the transition state." The energy required for the displacement...

General improvements and editing should be carried out to improve the readability.

MTD

Reviewer 1

1. The authors compared several bimetallic catalysts in PDH. However, only the reactant conversion and propylene selectivity were compared. In my view point, most bimetallic and trimetallic catalysts suffer from dealloying due to the repeated redox cycles during a regeneration process. I strongly recommend the authors to additionally compare the recyclability of the prepared catalysts under a fixed regeneration condition. I believe that the present Pb-PdGa catalyst would be superior to other catalysts.

Reply: Thank you for the important advice. Considering the reviewer's comment 2, we first tested the effect of O₂ concentration (5% and 20% O₂/He), then compared the recyclability of several catalysts. For both O₂ concentration, the catalytic performance of the PtGa-Pb catalyst recovered after some induction period (Figure S18), indicating that aerobic oxidation is also available for the regeneration of PtGa-Pb/SiO₂. However, other bimetallic or trimetallic Sn-containing catalysts (Pt₃Sn/SiO₂, PtSn/SiO₂, and PtGa-Sn/SiO₂) did not recover the original catalytic activity (Figure S19). Therefore, as the reviewer suggested, the recyclability of the PtGa-Pb/SiO₂ catalyst is superior to those of other catalysts.

We modified the sentences as follows: (p. 9, line 25) "We also tested the recyclability of PtGa-Pb catalyst (Figure S18). The spent PtGa-Pb catalyst could be regenerated by O₂ treatment to recover the original catalytic performance after some induction period, while some other bimetallic or trimetallic Sn-containing catalysts (Pt₃Sn, PtSn, and PtGa-Sn) did not (Figure S19)."

Figure S18. Catalytic performances of PtGa-Pb/SiO₂ (Pt/Pb = 2) at 650°C before (first run) and after (second run) regeneration. The regeneration process was carried out under flowing (a) 5% O₂/He ($F = 50 \text{ mL min}^{-1}$) at 550°C for 5 h and (b) 20% O₂/He ($F = 50 \text{ mL min}^{-1}$) at 550°C for 1 h, and the subsequent H₂ reduction at 650°C for 0.5 h.

Figure S19. Effect of regeneration process for the dehydrogenation of propane on (a) Pt₃Sn/SiO₂, (b) PtSn/SiO₂, and (c) PtGa-Sn/SiO₂ (Pt/Sn = 2) catalysts at 650°C. The regeneration process was carried out under flowing 20% O₂/He ($F = 50 \text{ mL min}^{-1}$) at 550°C for 1 h and the subsequent H₂ reduction at 650°C for 0.5 h.

Besides, the following sentences were added in Supporting information for readers' help: (pS23 in SI, line 2) “We tested the recyclability of the prepared catalyst. Prior to the second catalytic run, the spent catalysts were calcined under flowing O₂/He (5% for 5 h or 20% for 1 h) at 550°C and subsequently reduced under flowing H₂ at 650°C for 0.5 h. We chose the calcination temperature of 550°C according to literature of a Pt–Ga system, where Pt dispersion drastically decreased after oxidation at 650°C, whereas it retained at 550°C.³⁹ This result indicates that oxidation at 650°C results in some irreversible structural changes (probably, severe oxidative dealloying or sintering). Considering our TPO experiments shown in Figure S14, the coke accumulated on the spent catalysts was completely combusted at 550°C, demonstrating that the oxidation temperature of 550°C is high enough for coke removal. Figure S18a shows the changes of catalytic performance of PtGa-Pb/SiO₂ catalyst before and after a regeneration process with 5% O₂. Upon the regeneration, the catalytic performance was recovered to the original level after a short induction period. This induction period might be attributed to catalyst reconstruction: the catalyst structure (possibly, the placement of Pb atoms at the surface of PtGa) may be partly rebuilt during the regeneration process and further reconstruction to the original state occurs during the initial state of the 2nd catalytic run. We also tested the different calcination condition under 20% O₂/He (as a model for aerobic oxidation) for PtGa-Pb, which showed a similar trend with a longer induction period (Figure S18b). Thus, the catalyst can be reused by the simple regeneration procedure and has high resistance against the sintering of nanoparticles, as reported for Pt–Ga bimetallic systems.^{40,41} The sintering of nanoparticles is one of the biggest issues in the dehydrogenation of propane.¹³ On the other hand, other Sn-containing bimetallic (Pt₃Sn and PtSn) and trimetallic (PtGa-Sn) catalysts did not recover the original catalytic performances after the regeneration process (Figure S19). Particularly for PtSn, the C₃H₈ conversion and C₃H₆ selectivity significantly dropped. Thus, the PtGa-Pb/SiO₂ catalyst has a better recyclability than the Sn-based catalysts.”

2. The authors seemed to use carefully controlled regeneration conditions; oxidation under 5% O₂/He at 550°C followed by H₂ reduction at 650°C. Is there specific reasons for choosing such a condition? What will happen if the coked catalyst is simply calcined in air (20% O₂) at the reaction temperature (more practical conditions)?

Reply: Thank for the important comment. This is simply because we wanted to prevent severe oxidative dealloying, which we anticipated to occur at higher temperature and O₂ concentration. However, we have not optimized the O₂ concentration; honestly speaking, the 5% concentration was determined by intuition. Anyway, as noted in the reply for comment 1, the catalyst could be regenerated also under 20% O₂/He (but with a longer induction period).

For the temperature of oxidation, we chose 550°C according to the suggestion in literature of a Pt–Ga system (*Angew. Chem. Int. Ed.*, **2014**, 53, 9251): Pt dispersion drastically decreased after oxidation at 650°C, whereas it retained at 550°C. This result indicates that the higher oxidation temperature (650°C) results in some irreversible structural changes (probably, severe oxidative dealloying or sintering). Our TPO profiles (Figure S14) revealed that the coke accumulated on the spent catalysts was completely combusted at 550°C, demonstrating that the oxidation temperature of 550°C is high enough for coke removal. These comments were added in the supporting information as noted in the reply for comment 1.

3. I am not sure whether the catalyst can be described as a “single-atom” catalyst. The Pb-modified PtGa catalyst does not have exposed Pt ensembles due to the site blocking by Ga and Pb. However, “single-atom catalysts” generally mean the metal atoms or ions isolated within a “support” matrix.

Reply: Yes, as the reviewer pointed out, single-atom catalysts generally mean the metal atoms or ions isolated within a support matrix. On the (111) plane of intermetallic PtGa, the isolated Pt atom is placed on the Ga₃ trimer. We here regard the Ga₃ trimer as a “matrix” in a broad sense to support single-atom Pt. A similar description can be seen in literature on intermetallic PdGa: “single atoms (Pd₁) and trimers (Pd₃)” (*J. Phys. Chem. C*, 2014, 118, 12260). Of course, we are aware that this description is not common; therefore, we have described the active sites as “single-atom-like isolated Pt” mostly in the manuscript, except in the title and abstract. However, we would like to have this opportunity to propose that the description “single-atom Pt” can be used also for the case presented in this study, so that the field of single-atom chemistry is extended. This is why we used this description in the title and abstract. To make it clear, the following sentence was added: (p4, line 10) “Here, the Ga₃ moiety can be regarded as a matrix to support the isolated Pt₁ atom; therefore, it may be possible to describe the Pt₁ site as “single-atom Pt”.”, (p4, line 13) “In an analogous system of PdGa (space group *P*2₁3), such surface termination (Pd₃ and Pd₁, which were described as trimer and single atom, respectively) has actually been observed by surface science techniques.^{21,22}”.

Reviewer 2

1. pg 3 line 6: “Owing to the endothermicity and limited equilibrium propane conversion, high temperatures are requires...” – limited equilibrium propane conversion does not make sense.

Reply: According the comment, we deleted the words “limited equilibrium propane conversion”.

2. pg 3 line 16: “For Pt-based catalysts, Pt–Pt ensembles are known to be active for deep C–H(C) activation and their hydrogenolysis.” – what is “deep” C-H activation? Is it C-H scission from lower valency C atoms? The term is used in several places and is not, as far as I’m aware, commonplace in the C-H activation literature.

Reply: What we wanted to represent is “over-dehydrogenation”. We modified the sentence as follows (p3, line 16); “For Pt-based catalysts, Pt–Pt ensembles are known to be active for over-dehydrogenation of propylene and its hydrogenolysis.”

3. Figure 2d: A more appropriate label for the x-axis would be “Sputtering Cycle”.

Reply: Thank you for the suggestion. The label was modified as suggested.

4. pg 6 line 20: “CO pulse chemisorption experiment supported the partial coverage of surface Pt, where Pt dispersion decreased from 9.9% to 5.9% upon the Pb modification to PtGa/SiO₂” – this isn’t terribly clear though after checking Table S11, it seems that as Pt/Pb decreases, the Pt % dispersion also decreases. The text should be revised to convey this. Also, the “Pt dispersion” should be defined.

Reply: In this work, the loading amount of Pt was fixed at 3 wt% for all catalysts (please see the experimental details for Catalyst Preparation in Supporting information). Therefore, the decrease in Pt/Pb ratio just indicates the increase in Pb amount on the catalyst. To make it clear, the loading amount of Pt was noted in Table S1 caption as follows; “The loading amount of Pt is fixed at 3 wt% for all the catalysts.”. The definition of Pt dispersion was added as follows: (pS3 in SI, line 28) “The dispersion of Pt (percentage of exposed Pt to the total amount of Pt) in the catalysts was estimated by...”.

5. pg 8 line 19: “Although PtGa exhibited high conversion and selectivity at the initial stage (40% conv. 99.1% sel. at 0.5 h), conversion gradually decreased below half within 50 h.” – half of it’s initial value, rather than absolutely half.

Reply: According to the comment, we modified the sentence as follows; “Although PtGa exhibited high conversion and selectivity at the initial stage (40% conv. 99.1% sel. at 0.5 h), conversion gradually decreased below half of its initial value within 50 h.”

6. pg 11 Figure 5 and discussion: units of eV are used in the figure and mostly kJ/mol in the text/Table 1. This should be made uniform.

Reply: Thank you for pointing out. The units in Figure 5 and discussion were integrated into kJ/mol.

7. pg 11 line 15: “The displacement of the hydrogen atom to be eliminated becomes larger due to the absence of closely neighboring Pt, which significantly destabilizes the transition state.” The energy required for the displacement... General improvements and editing should be carried out to improve the readability.

Reply: We are sorry for the confusing description. What we wanted to say is as follows (p. 11, line 14); “Because of the molecular rotation and long Pt–Pt distance between the Pt₁ site and the nearest neighboring Pt₃ site (3.06 Å), the hydrogen atom involved in the third C–H scission has to migrate a long distance toward

the final state. The energy required for such an unfavorable path becomes significantly high.” We modified the description like this.

REVIEWERS' COMMENTS:

Reviewer #1 (Remarks to the Author):

I believe that the authors carefully addressed all of my comments and now the paper is ready for acceptance.